# The Influence of Rolling Direction and Dynamic Strengthening on the Properties of Steel

**DOI:** 10.3390/ma18163808

**Published:** 2025-08-13

**Authors:** Jakub Pokropek, Janusz Kluczyński, Bartłomiej Sarzyński, Katarzyna Jasik, Ireneusz Szachogłuchowicz, Jakub Łuszczek, Zdeněk Joska, Marcin Małek, Janusz Torzewski

**Affiliations:** 1Institute of Robots & Machine Design, Faculty of Mechanical Engineering, Military University of Technology, 2 Gen. S. Kaliskiego St., 00-908 Warsaw, Poland; jakub.pokropek@student.wat.edu.pl (J.P.); bartlomiej.sarzynski@wat.edu.pl (B.S.); katarzyna.jasik@wat.edu.pl (K.J.); ireneusz.szachogluchowicz@wat.edu.pl (I.S.); jakub.luszczek@wat.edu.pl (J.Ł.); janusz.torzewski@wat.edu.pl (J.T.); 2Department of Mechanical Engineering, Faculty of Military Technology, University of Defence, 662 10 Brno, Czech Republic; zdenek.joska@unob.cz; 3Faculty of Civil Engineering and Geodesy, Military University of Technology, 2 Gen. S. Kaliskiego St., 00-908 Warsaw, Poland; marcin.malek@wat.edu.pl

**Keywords:** ST700MC steel, tensile strength, digital image correlation, dynamic load, mechanical engineering

## Abstract

The article presents an analysis of the mechanical properties of S700MC steel, which represents advanced low-alloy high-strength steels. The influence of microstructure, shaped by a controlled thermo-mechanical rolling process, on the strength, ductility, and resistance to cracking and fatigue of the material is discussed. Particular attention is paid to the anisotropy of mechanical properties resulting from the orientation relative to the rolling direction, manifested by variations in yield strength, tensile strength, and total elongation of the specimens. The analysis also includes the material’s behavior under dynamic conditions, where the steel’s strength increases with the strain rate. Experimental investigations conducted using the digital image correlation (DIC) method enabled a detailed assessment of local strains and fracture characteristics of specimens subjected to both static and dynamic testing. The results showed that specimens cut along the rolling direction exhibited, on average, 6.4% higher tensile strength and 6.8% higher yield strength compared to those cut transversely. Moreover, dynamic loading led to an increase in load-bearing capacity of over 10% compared to static tests. The obtained data are highly relevant from the perspective of structural design, where the selection of material orientation and the consideration of strain rate effects are crucial for ensuring the reliability of components made from S700MC steel.

## 1. Introduction

Contemporary requirements in materials engineering, particularly in the transportation, energy, and construction sectors, impose the necessity for structural materials to simultaneously provide high mechanical strength, adequate ductility, and good weldability, as well as resistance to cracking and fatigue [1,2]. In response to these demands, high-strength low-alloy steels (HSLAs) have been developed, whose properties result from both their chemical composition (microalloying elements, such as Ti, Nb, and V) and technological processes, like rolling [3,4]. A notable representative of this material class is S700MC steel, widely used in the manufacture of structural components subjected to complex stress states and significant dynamic loads, such as vehicle frames, agricultural machinery parts, lifting equipment, and passive safety systems (e.g., road barriers) [5,6,7,8,9].

S700MC steel is characterized by a yield strength exceeding 700 MPa, high fracture resistance, and good cold formability. The microstructure of this material primarily consists of fine-grained ferrite with a bainitic phase, achieved through thermo-mechanical controlled processing (TMCP) and additions of Nb, Ti, and V. The presence of these elements promotes the formation of stable carbides, which effectively impede dislocation movement and enhance mechanical properties via dispersion strengthening [9,10,11,12].

The literature extensively documents the benefits of employing S700MC steel in lightweight structures. Vuorinen et al. [12,13] demonstrated that this material can achieve tensile strengths on the order of 1000 MPa while maintaining good impact toughness (approximately 150 MJ/m^3^), making it competitive with traditional structural steels, such as S355JR. The reduction in the mass of steel components by using S700MC directly translates into fuel consumption savings in the automotive sector—it is estimated that a 10% reduction in vehicle weight can lead to a 6–8% decrease in fuel consumption.

Despite numerous advantages, the steel exhibits anisotropy in its mechanical properties, resulting from the development of specific crystallographic textures due to rolling. Kovács [13,14] showed that the specimen’s orientation relative to the rolling direction significantly affects yield strength, tensile strength, and total elongation. Specimens cut transverse to the rolling direction demonstrate higher strength but lower ductility than longitudinal specimens. These effects are directly related to grain orientation and stress accumulation along the principal deformation axis.

The mechanical response of S700MC steel under dynamic loading conditions is also sensitive to anisotropy and requires separate analysis. HSLA steels exhibit positive strain rate sensitivity—their strength increases with the rate of loading, which is explained by the activation of strain rate hardening mechanisms and restricted dislocation mobility. Constitutive models, such as the Johnson–Cook model, are often used for numerical simulation of steel behavior at high strain rates, although recent studies indicate the need to modify these models by incorporating dynamic softening effects (e.g., recrystallization) in the deformation analysis of microstructurally complex metals [14,15,16,17,18].

The controlled rolling process plays a key role in shaping the microstructure and mechanical properties of HSLA steels. Hot rolling involves precise control of temperature and deformation during sequential thermo-mechanical processing stages, enabling the formation of a fine-grained structure and inhibiting austenite grain growth prior to transformation into ferrite and bainite. Oktay et al. [18,19] demonstrated that at a coiling temperature of approximately 450 °C, the microstructure comprises ferrite and bainite. When the temperature rises to about 600 °C, a ferrite–carbide structure with martensite presence emerges. This leads to an increase in tensile strength by at least 80 MPa, although it results in reduced impact toughness and a more continuous yield point phenomenon.

A review of the available literature and material characterization indicates that S700MC steel belongs to the group of advanced high-strength low-alloy steels (HSLAs), whose mechanical properties derive from a precisely controlled thermo-mechanical rolling process and the addition of microalloying elements, such as niobium, vanadium, and titanium. The rolling procedure critically influences the final microstructure—fine-grained ferrite with bainitic admixture—by promoting the formation of stable carbides that hinder dislocation movement. A significant effect of rolling direction on the anisotropy of mechanical properties has also been observed, manifested in differences in yield strength and total elongation depending on the orientation relative to the rolled sheet. Against this background, the aim of the present article is an experimental assessment of the influence of rolling direction and dynamic loading conditions on the mechanical properties of S700MC steel, with a particular emphasis on strength parameters, local strain phenomena analyzed by digital image correlation (DIC), and fracture characteristics of specimens subjected to static and dynamic tests. Despite the extensively documented properties of S700MC steel, there is still a lack of experimental data combining the effects of rolling direction and strain rate on mechanical behavior, analyzed simultaneously using DIC. In particular, few studies present a quantitative analysis of anisotropy under both quasi-static and dynamic conditions, accompanied by the observation of local strain fields.

The primary objective of this study is to quantitatively assess the influence of rolling direction and dynamic loading on the mechanical properties of S700MC steel using the DIC method. This combination of static and dynamic experiments enabled not only the identification of material anisotropy but also the analysis of local deformation phenomena, including necking and crack propagation. The conducted research provides valuable data for engineering modeling of S700MC components under service conditions involving variable stress orientations and high strain rates.

## 2. Materials and Methods

To verify the influence of rolling direction and dynamic loading on the mechanical properties of S700MC steel, a series of tests was designed, including static and dynamic tensile tests, specimen geometry measurements, microstructural analysis, and fracture documentation. The final element of the experimental procedure was digital image correlation (DIC), which enabled the recording of local strain fields and the identification of damage initiation sites. This section provides a detailed description of the materials used, the experimental procedures, and the measurement equipment.

### 2.1. Material—S700MC Steel

The material used in the study was S700MC steel (also referred to in some publications as S700). This is a high-strength structural steel widely employed in the automotive and construction industries due to its favorable mechanical properties and good formability. The material was manufactured and delivered in accordance with the requirements of the EN 10083 standard, ensuring consistent quality and compliance with international specifications [20]. Specimens for mechanical testing and final component fabrication were cut from a sheet measuring 1500 × 3000 mm with a thickness of 3 mm. Prior to cutting, a preliminary surface and dimensional inspection was carried out to eliminate the presence of any material defects that could affect test outcomes. Table 1 presents the chemical composition and basic mechanical properties as provided by the steel manufacturer, ThyssenKrupp AG (ThyssenKrupp AG, Essen/Duisburg, Germany), one of Europe’s leading suppliers of specialized steels.

This steel is characterized by a relatively high content of manganese and molybdenum, which enhance strength and creep resistance, along with a low carbon content (up to 0.06%), which positively influences the material’s weldability. The tensile strength of the material ranges from 750 to 900 MPa, with a yield strength of approximately 700 MPa. The elongation after fracture, measured on a specimen with a gauge length five times the diameter (A5), is 16%, while elongation measured over an 80 mm gauge length is 12%, indicating moderate ductility while maintaining high strength.

### 2.2. Microstructure Investigations

According to the information provided by the manufacturer, the microstructure of the material is primarily ferritic, containing nanoscale precipitates that contribute to additional strengthening through dispersion hardening mechanisms. This microstructural configuration offers a favorable balance of mechanical and technological properties, enabling the material to perform effectively under conditions requiring both high strength and resistance to permanent deformation [21]. To verify the microstructure, a metallographic specimen was prepared. After grinding and polishing, the specimen was etched using a 2% Nital solution. Microstructural images were captured using an Olympus 4000 LEXT laser confocal microscope (Olympus, Tokyo, Japan).

### 2.3. Samples for Tensile Test

All tests were conducted on specimens prepared in accordance with the ASTM E8/E8M-2016a standard [22]. The shape and dimensions of the test specimens are shown in Figure 1. The specimens had an overall length of 100 mm and a cross-sectional area of 3 × 6 mm in the gauge section.

Due to the anisotropy of mechanical properties depending on the rolling direction, and in order to verify the manufacturer’s declared mechanical characteristics, tensile test specimens were prepared in two orientations, namely parallel and perpendicular to the rolling direction. This approach enables the assessment of material homogeneity and the identification of any deviations from the specified parameters. A schematic illustration of the specimen extraction layout in relation to the sheet metal is presented in Figure 2.

To ensure repeatability of the results, five specimens were prepared for each tested configuration. Prior to cutting, the sheet was carefully measured and cutting lines were marked to eliminate dimensional inaccuracies. The specimens were cut from the sheet metal using a guillotine and a band saw. The final contour was shaped using a CNC milling machine, ensuring high dimensional precision and consistency of specimen geometry. During all machining operations, a coolant–lubricant spray was applied to minimize tool wear and reduce the thermal impact of cutting on the material’s structure and mechanical properties. In addition, the temperature of the machined material was monitored immediately after processing to ensure that it did not exceed levels that could alter its mechanical characteristics. Figure 3 shows the first batch of specimens prepared for static tensile testing. To facilitate identification during mechanical testing and data analysis, specimens were labeled accordingly. Samples cut perpendicular to the rolling direction were marked as SX1, SX2, SX3, SX4, and SX5. Samples cut parallel to the rolling direction were labeled as SY1, SY2, SY3, SY4, and SY5.

### 2.4. Cross-Section Measurements

After the fabrication of all specimens, the cross-sectional area of the gauge section was measured for each sample. This step allowed verification of the dimensional accuracy achieved through the machining process. Measurements were taken at three different locations along the gauge length, and the arithmetic mean along with the standard deviation was calculated. The measurement points are illustrated in Figure 4. Measurements were performed along the specimen’s axis of symmetry, as well as 10 mm to either side. The individual measurement points were labeled D1, D2, and D3. Using the average values, and applying Equation (1), the cross-sectional area for each specimen was calculated as follows:(1)a = t · w
where

a—cross-sectional area (mm^2^);

t—average thickness (mm);

w—average width (mm).

**Figure 4 materials-18-03808-f004:**
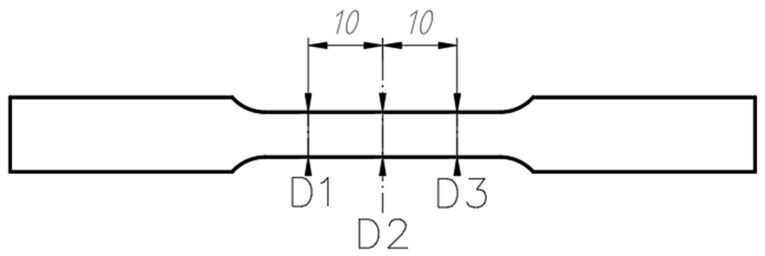
Position of measurement points relative to the specimen (point D2 lies on the specimen’s axis of symmetry).

### 2.5. Tensile Tests and DIC Measurements

The tensile strength tests of the specimens were carried out using the Instron 8802 servo-hydraulic testing system (Norwood, MA, USA). This system is designed for mechanical testing under loads up to 250 kN. To measure axial strain along the direction of applied force, an Epsilon 3542 extensometer (Epsilon Technology Corp., Jackson, WY, USA) was employed, capable of measuring axial strains in the range of 0 to 10%. Additionally, a DIC system from Dantec Dynamics (Dantec, Ulm, Germany) was used to provide a non-contact method for tracking strain development during the tensile tests. The DIC technique involves capturing a sequence of images throughout the loading process and analyzing them to detect deformations based on a random speckle pattern applied to the surface of the specimen. By comparing successive images, the system calculates local displacements and strain fields with high precision. This method allows for a full two-dimensional strain distribution to be obtained in real time. To facilitate accurate DIC analysis, specimens were marked with randomly distributed paint speckles. All tests were performed in accordance with the ASTM E8/E8M-2016a, standard test methods for tension testing of metallic materials [22]. During testing, each specimen was mounted in the grips of the tensile testing machine by its gripping section. The specimens were subjected to tensile loading until either surface rupture occurred or an elongation of 5% was reached, whichever came first. The applied strain rate was 0.01 mm/min. Figure 5 presents the tensile testing machine used and an image captured during the tensile test, including the DIC system analysis. The use of both extensometry and DIC enabled cross-verification of strain data and increased measurement accuracy, especially near the yield point. This dual-measurement approach also allowed for early detection of localized necking prior to final fracture. The synchronization between mechanical load data and optical imaging provided a comprehensive understanding of deformation behavior across different regions of the specimen.

### 2.6. Fracture Analysis

Fracture surfaces of the specimens after the static tensile test were imaged using a KEYENCE VHX-7000 digital microscope (Keyence, Osaka, Japan). Analysis of the fracture surfaces enables identification of the mechanisms occurring during the strength test, as well as characterization of the material and the crack propagation path.

## 3. Results and Discussion

### 3.1. Microstructure Analysis

The microstructure of S700MC steel was analyzed using a laser confocal microscope (Olympus, Tokyo, Japan).. Observations revealed a homogeneous, fine-grained structure typical of thermomechanically processed steels. The dominant phase was ferrite, with the presence of bainite and fine non-metallic inclusions. The examined samples exhibited very fine, uniformly distributed grains with a regular shape, indicating effective controlled rolling during processing. No significant discontinuities or defects were observed in the microstructure, confirming the high quality of the material. Some images (Figure 6) display a subtle texture resulting from directional deformation during rolling. Such a microstructure contributes to the material’s high mechanical strength and good plasticity, making S700MC steel well-suited for applications involving significant dynamic loads and plastic forming processes. Comparison with microstructural images available in the scientific literature confirms a high degree of consistency [7,23,24]. It should be noted that, in this study, metallographic analysis was limited to qualitative evaluation without quantitative grain size measurements or phase composition determination according to ASTM E112 [25]. The scope of the current work did not allow for this type of analysis; however, such an approach is planned in future research to establish a statistical correlation between microstructural parameters and anisotropic mechanical properties.

### 3.2. Cross-Section Area Measurements

To verify the correct fabrication of test specimens for the static tensile test, their diameters were measured at three different locations. Using the measured thickness and width values of the gauge section, the arithmetic mean was calculated. Based on the average values and Equation (1), the cross-sectional area of each tested specimen was determined. Standard deviations were also calculated for the obtained values. The results are presented in Table 2 and Table 3. The calculated standard deviations for thickness and width measurements ranged from 0.0058 mm to 0.0100 mm, which indicates that the variability of individual measurements is low. These values are considered small and are typical for manual or semi-automated caliper measurements under laboratory conditions. Therefore, they do not introduce significant uncertainty into the final calculated cross-sectional areas. The standard deviations of the cross-sectional area themselves ranged from approximately ±0.035 mm^2^ to ±0.063 mm^2^. Given that the average areas are close to 18.5 mm^2^, these uncertainties correspond to a relative uncertainty of only about 0.2–0.3%, which is minimal. Hence, it can be concluded that the uncertainty in cross-sectional area measurements has a negligible impact on the accuracy of the subsequent tensile strength calculations. The observed measurement uncertainties result primarily from slight operator inconsistencies in positioning the measuring tool (e.g., calipers), surface irregularities on the specimen, and material anisotropy introduced during rolling. In one case—specimen SX4—a higher standard deviation in area (±0.3510 mm^2^) was recorded, which is an outlier compared to the rest. This higher uncertainty is likely to result from local geometric imperfections, such as burrs or uneven surface finish, rather than from systemic error. This suggests that the fabrication process for this specimen may have been slightly less precise. Despite this, the absolute variation remains small relative to the area, and its impact on strength calculations is limited.

Among the SX specimens, the smallest cross-sectional area was recorded for specimen SX1, with a value of 18.3947 ± 0.039 mm^2^, while the largest was specimen SX4, with 18.9660 ± 0.0394 mm^2^. For the SY specimens, the smallest average cross-sectional area was found in specimen SY1 (18.1774 ± 0.0389 mm^2^), and the largest was found in specimen SY4 (18.7347 ± 0.0632 mm^2^). For all specimens, the cross-sectional area ranged between 18 and 19 mm^2^, confirming the precision and consistency of the specimen fabrication process. All specimens showed cross-sectional areas within the expected range of 18–19 mm^2^, with relatively tight uncertainties, which confirms both the accuracy and repeatability of the measurement process. It also validates the uniformity of the specimen manufactured procedure, which is critical for ensuring the comparability of tensile test results.

### 3.3. Tensile Tests and DIC Measurements

During the strength tests, sensors and software integrated with the testing machine were used, allowing for real-time data acquisition and determination of key material properties. The applied force induced tensile stress in the specimens. A displacement sensor enabled the measurement of specimen strain at the moment of peak force application. The tensile test machine is equipped with a calibrated load cell and extensometer system. The uncertainty of force measurement, according to the manufacturer’s specification, is typically ±0.5% of the indicated value, while strain measurement using the extensometer system carries an uncertainty of approximately ±0.01%. These values are considered low and consistent with industry standards for mechanical testing.

Numerical data for specimens cut perpendicular to the rolling direction (SX) are presented in Table 4, while those cut parallel to the rolling direction (SY) are provided in Table 5. Using the testing machine’s software (BluehillUniversal Software, Instron, Norwood, MA, USA, version 3) the upper yield strength (ReH), ultimate tensile strength (Rm), and their corresponding strains (εReH, εRm) were recorded. Standard deviations were calculated for each mechanical property across the tested groups. The low dispersion of R_eH_ (±9.55 MPa for SX and ±11.56 MPa for SY) and R_m_ (±10.28 MPa for SX and ±9.93 MPa for SY) suggests high test repeatability. Relative uncertainty in the stress values is below 1.5%, which confirms the robustness of the test procedure. Similarly, strain measurements (εR_eH_, εR_m_) showed very low standard deviations (0.02–0.1%), indicating consistent elongation behavior among specimens. To better visualize the mechanical behavior of the material under load, engineering stress–strain curves were generated based on data from the testing machine. These are shown in Figure 7 and Figure 8. Analysis of the graphs revealed the presence of a distinct upper yield point for the material. Among the SX specimens, the highest recorded ultimate tensile strength was observed for specimen SX1, with a value of 781.48 MPa at a strain of 3.49%. This specimen also exhibited the highest upper yield strength, reaching 730.25 MPa at a strain of 0.41%. The lowest tensile strength was recorded for specimen SX4, which reached 757.75 MPa at a strain of 3.57%, with an upper yield strength of 706.42 MPa at a strain of 0.43%. Based on the test results for all SX specimens, the arithmetic means and standard deviations were calculated. The average upper yield strength was 716.36 ± 9.55 MPa at a strain of 0.40 ± 0.02%, while the average ultimate tensile strength was 768.04 ± 10.28 MPa at a strain of 3.60 ± 0.09%. Such low measurement dispersion confirms that the material exhibits homogeneous mechanical behavior under the tested conditions and that the experimental setup introduces minimal variability into the results.

Among the SY specimens, the highest recorded ultimate tensile strength was observed for specimen SY1, reaching 820.21 MPa at a strain of 3.07%. This specimen also exhibited the highest upper yield strength of all tested samples, at 781.01 MPa with a corresponding strain of 0.32%. The lowest tensile strength was recorded for specimen SY4, at 794.34 MPa and a strain of 3.24%, with an upper yield strength of 751.22 MPa at 0.32% strain. Based on the results from all SY specimens, the arithmetic means and standard deviations were calculated. The average upper yield strength was 765.11 ± 11.56 MPa at a strain of 0.36 ± 0.02%, while the average ultimate tensile strength was 805.61 ± 9.93 MPa at a strain of 3.16 ± 0.1%. The relatively higher scatter in R_eH_ for the SY group (±11.56 MPa) compared to SX may suggest slight microstructural differences or rolling-direction effects, yet the variation remains within acceptable limits for high-strength steels and has minimal influence on statistical significance. Analysis of the stress–strain curves presented in Figure 6 and Figure 7 shows that all specimens exhibit very similar behavior. In the elastic and elastic–plastic regions, the curves are nearly identical. Slight variations appear only at the onset of the upper yield point. The test specimens cut parallel to the rolling direction (SY) demonstrated significantly superior mechanical properties compared to those cut perpendicular to the rolling direction (SX). One possible reason for the increased strength in the SY specimens is the presence of a residual rolling texture, which often promotes higher dislocation resistance along the rolling direction. The elongated grain structure aligned with the load path may also contribute to improved load-bearing capacity and resistance to plastic deformation. Furthermore, the directional alignment of non-metallic inclusions—such as sulfides or oxides—may reduce their detrimental effect on crack initiation when oriented parallel to the loading axis. Although these factors were not directly measured in this study, they may partially explain the anisotropic behavior observed between SY and SX specimens. As a result, it was decided that final components should be cut in alignment with the rolling direction, which is expected to enhance the structural durability and reliability of the tested design It is important to note that in both test directions, the measured strengths (R_eH_ and R_m_) significantly exceeded the minimum values declared by the steel manufacturer, confirming both the quality of the material and the reliability of the experimental methodology.

To analyze material displacements under plane strain conditions during bending tests, the DIC method was employed. Figure 9 presents the measurement results for specimen SY4. The figure includes images of the specimen in five different states. The first image on the left shows the specimen prior to the static tensile test and is treated as the reference image for displacement analysis in subsequent deformation stages. To the right of the four deformation images, color scales are provided, representing the values of effective engineering strains. The DIC system introduces its own sources of uncertainty, including optical resolution limits, lighting conditions, lens distortion, and correlation errors. In this study, the DIC setup used a high-resolution camera and a validated calibration grid, achieving a displacement resolution on the order of ~0.01 mm and a strain resolution of approximately ±0.01%. Given that observed strains reach values above 3%, these uncertainties are negligible and allow for confident assessment of strain localization phenomena, such as necking. Based on the images—particularly those corresponding to the tensile strength and before crack states—a noticeable discrepancy between the stress–strain curve and the DIC measurement is observed. The stress–strain curve suggests a decrease in stress after reaching the tensile strength point. However, DIC analysis, consistent with the theory of engineering stresses, shows that after reaching the ultimate tensile strength, necking occurs and localized strain increases, which in turn causes the actual (true) stress in the material to continue rising. This divergence illustrates the limitations of conventional stress–strain representation and highlights the utility of DIC in providing spatially resolved strain data, particularly in post-yield and necking regions where standard extensometers cannot operate. The uncertainty of local strain data from DIC remains low enough to capture these nonlinearities effectively. Figure 10 presents the images recorded during the tensile test of specimen SX4. The individual images are similar in nature to those captured for specimen SY4. In the image corresponding to the stress level at the upper yield point, uniform deformation is observed across the surface of the specimen (indicated by a consistent color distribution). Compared to the reference image, a clear color shift is visible across the entire surface, indicating widespread strain. This uniform strain pattern suggests that the material responded predictably within the elastic and early plastic deformation ranges. The image captured at the point of maximum tensile stress shows the initiation of failure propagation—visible as orange/red regions, which correspond to the highest local stresses. These stress concentrations appear at the center of the specimen, consistent with theoretical expectations for necking in ductile metals. In the subsequent image, taken just before fracture, a well-defined necking zone is clearly visible, representing the region of maximum localized deformation. The onset of necking coincides with a plateau in the stress–strain curve, confirming the transition to unstable plastic flow. This zone corresponds to the actual true tensile strength of the material just before rupture, and the DIC analysis allows quantification of local strain fields with sub-millimeter precision. The uncertainty associated with this analysis is mainly linked to speckle pattern quality and image noise, both of which were minimized using appropriate surface preparation and calibration.

Although the DIC images effectively demonstrate localized strain accumulation and validate the onset of necking, it is important to note that true stress–strain curves were not calculated in this study, due to the lack of numerical full-field strain data. Nevertheless, based on the DIC observations, it can be inferred that while engineering stress appears to decrease after reaching the ultimate tensile strength, the actual (true) stress in the necking region continues to rise. This qualitative insight aligns with the theoretical distinction between engineering and true stress–strain behavior in ductile materials.

Dynamic tests involved subjecting the specimens to impact loading until failure. The objective was to evaluate the structural response of the material under sudden overload conditions, simulating scenarios, such as accidental impacts or high-speed collisions. These tests provided critical insights into the behavior of the material when subjected to rapid deformation rates, which differ significantly from those observed under quasi-static loading. Figure 11 presents the stress–strain curves recorded during the dynamic tensile test. A reference curve with SY1 data (highest Rm) was added.

The above graph presents a comparative stress–strain analysis under static and dynamic tensile testing conditions. Curves D1, D2, and D3 represent three dynamic test replicates, while SY1 reflects a quasi-static test. All samples demonstrate a linear elastic region followed by distinct yielding behavior; however, significant differences emerge in post-yield behavior between dynamic and static tests. When comparing the results from static and dynamic tensile tests, the ultimate tensile strength (Rm) increased by approximately 19% under dynamic loading. This result clearly demonstrates the strain-rate sensitivity of the investigated S700MC steel. Comparable behavior has been reported for other HSLA steels, where dynamic loading at similar strain rates resulted in tensile strength increases in the range of 6–20%, depending on alloy composition, microstructural state, and testing temperature [26,27,28,29]. The observed strengthening effect is consistent with mechanisms described in the literature, including the suppression of dislocation motion due to reduced time for dynamic recovery processes and the influence of microstructural features, such as grain refinement and precipitate distribution [30,31]. These findings confirm that the tested S700MC steel exhibits strain-rate sensitivity typical for its class, reinforcing its suitability for applications where components are exposed to impact or crash loads. From an engineering standpoint, this effect should be taken into account in the design of safety-critical structures, as the increased strength under dynamic loading may improve energy absorption capacity and delay the onset of catastrophic failure. In the case of the static sample (SY1), the curve exhibits the expected characteristics for ductile metals: after yielding (~750 MPa), the stress continues to increase moderately, reaching an ultimate tensile strength (UTS) of ~820 MPa. This is followed by a visible strain-hardening plateau and a gradual drop due to necking and eventual fracture. This behavior is typical for low to medium strain rates ($\overset{\cdot }{\varepsilon }$ < 10^−2^ s^−1^), where the material has sufficient time to undergo plastic deformation mechanisms, such as dislocation motion, multiplication, and work hardening. Conversely, the dynamic samples (D1–D3) exhibit a much steeper stress–strain path, with maximum stresses approaching ~970 MPa, followed by an abrupt stress drop with no apparent strain-hardening plateau or necking region. This response is characteristic of high-strain-rate tensile loading (e.g., $\overset{\cdot }{\varepsilon }$ > 100 s^−1^), where the material fails almost immediately after reaching UTS. The absence of necking and the brittle-like fracture indicate that the specimens fractured without localized deformation. The uniformity and repeatability of curves D1 through D3 reinforce the reliability of the observed behavior. The similar peak stress and immediate post-UTS drop confirm that the fracture was not influenced by sample preparation artifacts or test irregularities. Instead, it reveals a genuine material response to high-strain-rate tensile loading.

### 3.4. Fracture Analysis

After the completion of the static tensile test, specimen SX4 was subjected to a detailed fractographic analysis using optical and digital imaging techniques (Figure 12). This procedure aimed to investigate the fracture mechanisms at the microstructural level and verify whether the material exhibited typical failure modes associated with thermomechanically processed high-strength steels. Based on the observed fracture surface, the primary mechanisms leading to failure could be clearly identified and classified. The macroscopic appearance of the fracture indicates a combination of ductile and brittle features, suggesting a mixed-mode failure. The central region of the fracture surface shows the presence of internal discontinuities, marked with red ellipses, which may indicate microcracks or voids formed during rolling processes or as a result of stress accumulation under mechanical loading. These imperfections likely acted as stress concentrators, initiating localized plastic deformation that evolved into microvoid coalescence—a well-known ductile fracture mechanism. Their elongated geometry, oriented in the direction of tensile loading, provides direct evidence of directional crack propagation aligned with the principal stress axis. The morphology of the elongated cavities is consistent with plastic stretching under triaxial stress states, typically observed in the late stages of necking. A clearly defined necking region is visible in the upper part of the image, characterized by a significant reduction in cross-sectional area, which is indicative of intense localized plastic flow and is a hallmark of ductile fracture behavior. In the magnified images located at the bottom of the figure, features of the fracture surface become more discernible. Numerous micro-dimples and tear ridges are observed, representing the final rupture zone and crack initiation sites. These micro-dimples form due to the nucleation, growth, and coalescence of microvoids, which is typical for metals that undergo significant plastic deformation prior to fracture. Similar features have been observed in other HSLA steels, such as S500MC and S960QC, where the dominant ductile fracture mechanism involves microvoid formation around fine inclusions and second-phase particles [23,32]. The elongated shape of dimples and their alignment with the loading direction are also consistent with the literature reports describing void coalescence under triaxial stress states during necking. These micro-dimples and tear ridges are consistent with the fracture morphology described in HSLA steels, where quantitative analyses have been applied to dimple size and density under ductile tearing conditions. Their size and distribution can provide insight into the material’s toughness and damage tolerance. Red horizontal lines were used to mark the axis of symmetry of the specimen, helping to correlate the observed fracture features with the loading geometry. Additionally, the transition from ductile tearing in the center to smoother, shear-type zones near the edges suggests that the outer portions experienced different local stress states, possibly due to strain rate gradients or constraint effects. This further confirms the complex, multiaxial stress field developed during failure. Overall, the fracture surface exhibits mixed-mode characteristics, with a predominantly ductile nature and localized defects that could potentially reduce the material’s fatigue resistance. While the ductile region ensures that the material can absorb considerable energy before failure, the presence of microstructural anomalies indicates potential sites for early fatigue crack initiation in cyclic loading conditions, particularly in high-load structural applications. Therefore, improving material homogeneity and minimizing inclusion content could enhance long-term durability in service environments. The internal pores and microvoids observed in the fracture surface may originate from improper rolling practices, such as non-uniform deformation, segregation during casting, or insufficient inclusion removal. To minimize such microstructural defects, it is recommended to apply secondary metallurgical treatments (e.g., argon stirring or vacuum degassing), stricter control of chemical composition, and optimized coiling temperatures during TMCP.

## 4. Conclusions

As part of the conducted research, a detailed analysis of the mechanical properties of S700MC steel was carried out, with particular emphasis on the effect of rolling direction on the material’s strength. Experimental engineering methods were employed, including static tensile tests in accordance with ASTM E8/E8M-2016a, DIC, microstructural analysis, and fracture surface examination (fractography). The results enabled the evaluation of material homogeneity, the presence of mechanical anisotropy, and the identification of dominant deformation and fracture mechanisms. Based on the obtained data, the following conclusions were drawn:The rolling direction should be considered a key design parameter in structural elements made from S700MC steel. Aligning load paths with the rolling direction can improve tensile strength and reduce the risk of premature plastic deformation.In dynamic applications (e.g., crash scenarios, impact zones), S700MC exhibits an over 10% increase in strength. However, this is accompanied by a reduction in ductility and a brittle-like fracture, which should be accounted for in safety margin calculations.Microstructural defects, such as internal pores from rolling, may act as crack initiation sites. Their minimization during production is essential for components subjected to cyclic or high-rate loading.The use of DIC in design validation can support the detection of local strain concentrations and should be considered in advanced prototyping of safety-critical components.

In conclusion, the conducted investigations demonstrate that S700MC steel is a material with excellent mechanical properties, exhibiting predictable and favorable behavior under load, which makes it highly suitable for structural applications. The applied research methods allowed not only for quantitative strength assessment but also for an in-depth analysis of local deformation phenomena, making the findings a valuable contribution to the advancement of high-strength steel component design.

Although the obtained results confirm the favorable properties of S700MC steel, it should be noted that the study was preliminary in nature and involved a limited number of tests, including only a single dynamic experiment. Advanced crystallographic analysis and constitutive modeling for predicting material behavior under real-world conditions were also not included. Therefore, the interpretation of the results should take these limitations into account, and future research should involve fatigue testing, numerical simulations, and extended microstructural characterization.

## Figures and Tables

**Figure 1 materials-18-03808-f001:**
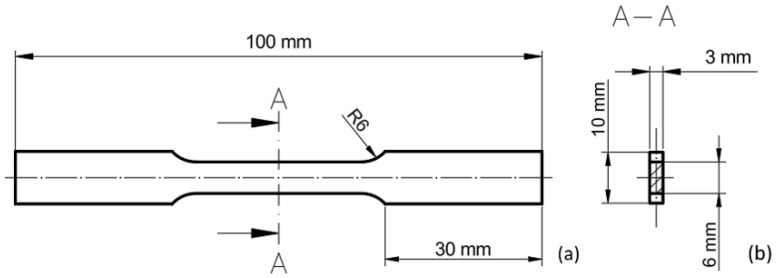
Geometry and dimensions of test specimens intended for static tensile testing. (**a**) Sample contour; (**b**) cross-section of measured area [22].

**Figure 2 materials-18-03808-f002:**
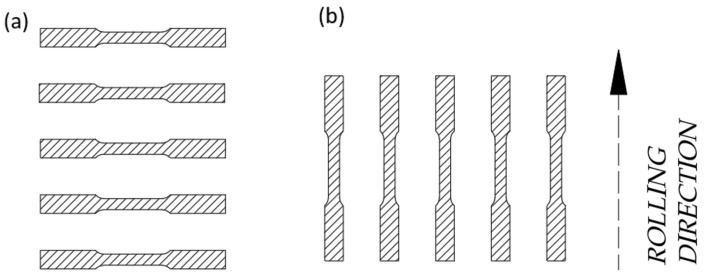
Orientation of specimens in relation to the rolling direction of the sheet metal. (**a**) Perpendicular to the rolling direction; (**b**) parallel to the rolling direction.

**Figure 3 materials-18-03808-f003:**
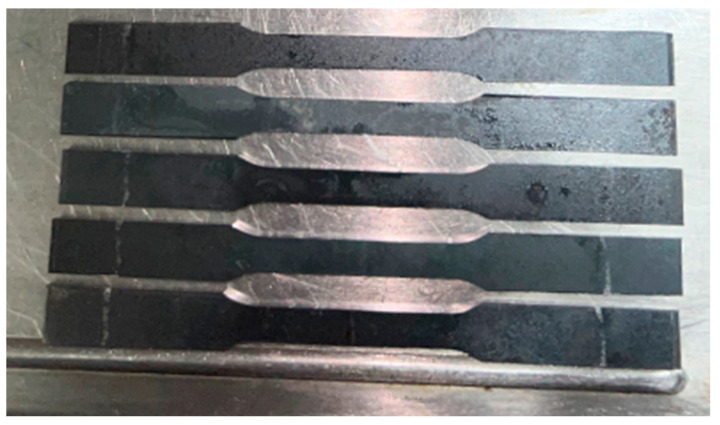
Specimens prepared for the static tensile test.

**Figure 5 materials-18-03808-f005:**
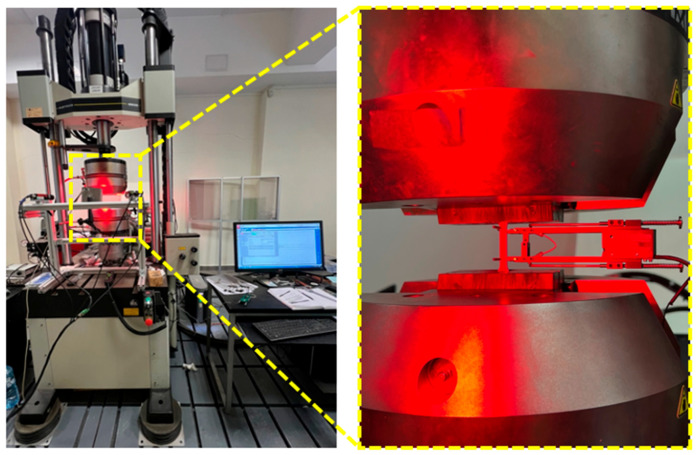
The Instron 8802 universal testing machine during a tensile test conducted with the use of the DIC system.

**Figure 6 materials-18-03808-f006:**
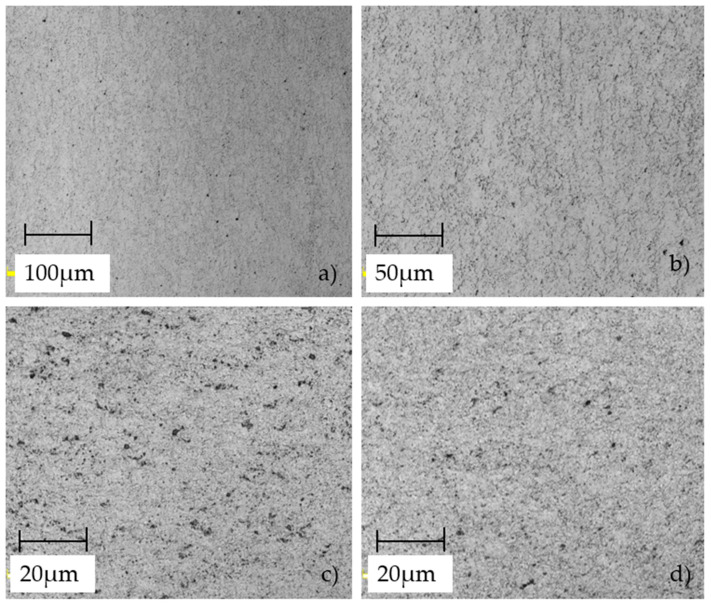
Images of the S700MC steel microstructure after hot rolling taken at different magnifications: (**a**) ×20, (**b**) ×50, and (**c**,**d**) ×100.

**Figure 7 materials-18-03808-f007:**
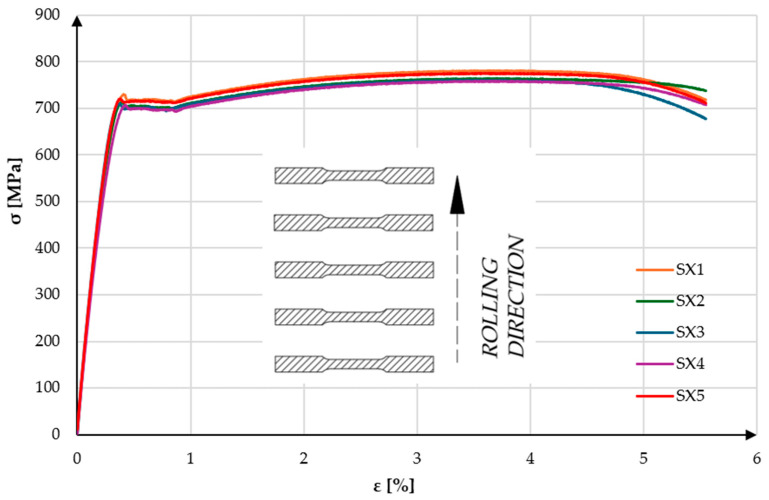
Stress–strain curves of the “SX” specimens obtained during a static tensile test.

**Figure 8 materials-18-03808-f008:**
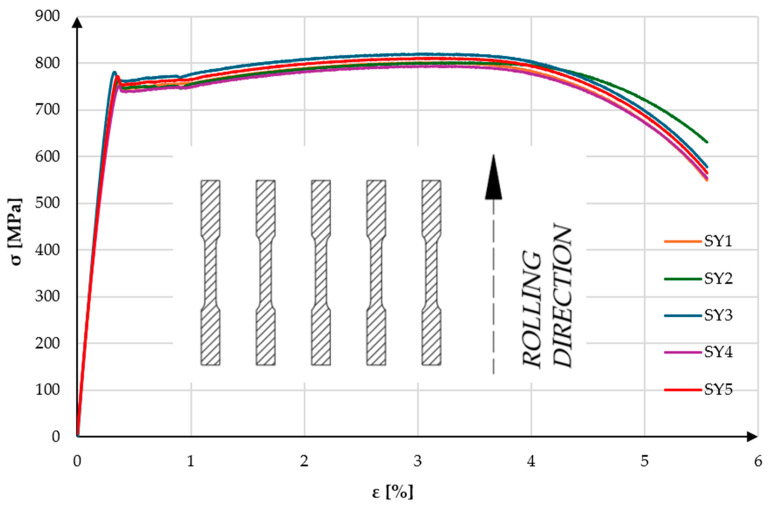
Stress–strain curves of the “SY” specimens obtained during a static tensile test.

**Figure 9 materials-18-03808-f009:**
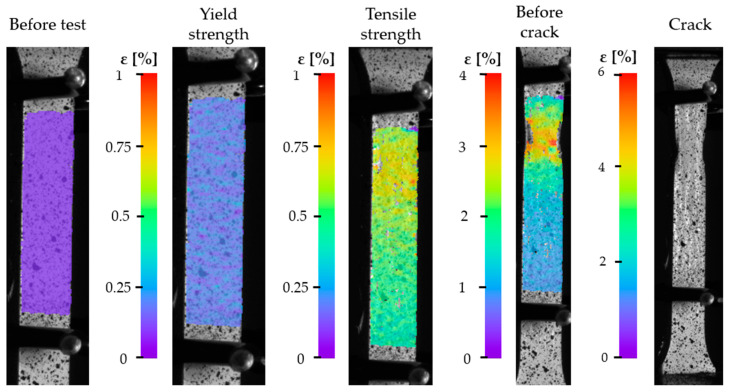
SY4 specimen under static tensile loading, captured using the DIC system.

**Figure 10 materials-18-03808-f010:**
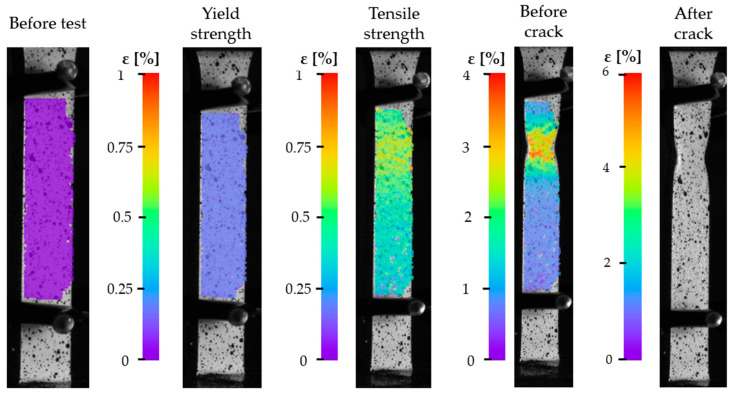
SX4 specimen under static tensile loading, captured using the DIC system.

**Figure 11 materials-18-03808-f011:**
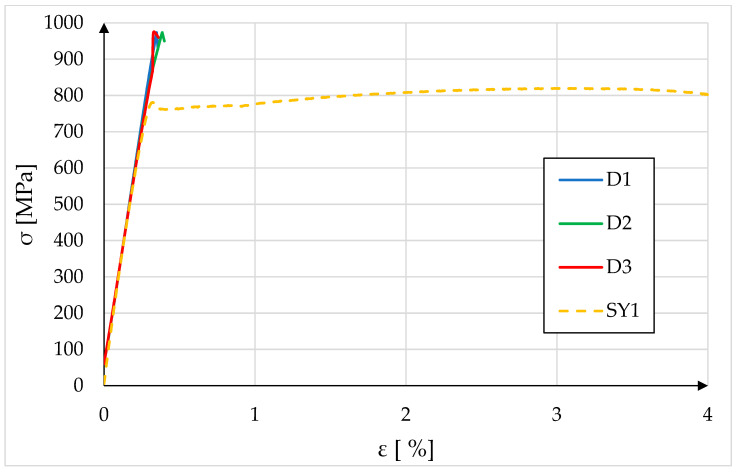
Results of dynamic tests and reference sample from the static tensile test vs. strain graph.

**Figure 12 materials-18-03808-f012:**
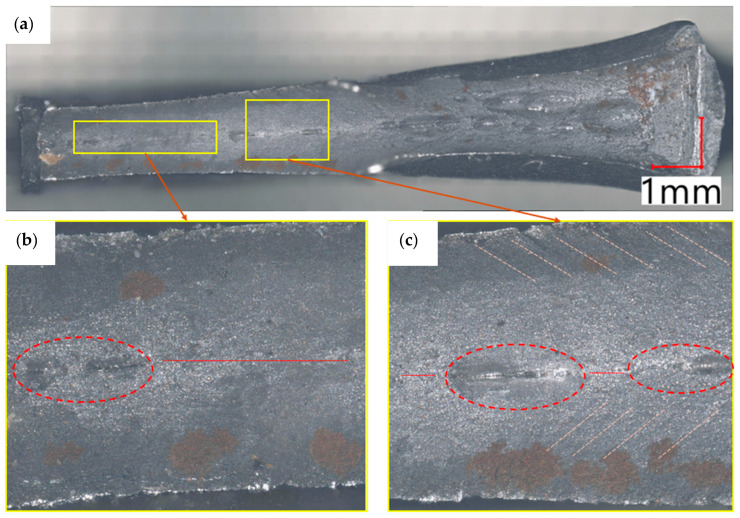
Photograph of the fracture surface of the SX4 sample, illustrating the regions of crack propagation: (**a**) sample integrals view; (**b**,**c**) regions of crack propagation.

**Table 1 materials-18-03808-t001:** Chemical composition and mechanical properties of S700 steel [21].

Element	Weight [%]
C	<0.06
Si	<0.15
Mn	<1.80
P	<0.020
S	<0.008
Al	0.015–0.060
Ti	<0.15
Nb	<0.08
V	<0.05
Mo	<0.30
B	<0.005
**Property**	**-**
Yield strength R_eH_ [MPa]	700
Tensile strength R_m_ [MPa]	750–900
Elongation [A_5_]	16
Elongation [A_80_]	12

**Table 2 materials-18-03808-t002:** Results of cross-sectional area measurements of the tested part of SX samples.

Test Sample	Measuring Point	Thickness [mm]	Average Thickness “t” [mm]	Standard Deviation [mm]	Width [mm]	Average Width “w” [mm]	Standard Deviation [mm]	Cross-Sectional Area “a” [mm^2^]	Standard Deviation [mm^2^]
SX1	D1	3.05	3.047	0.0058	6.04	6.037	0.0058	18.3947	0.0390
	D2	3.05			6.04				
	D3	3.05			6.03				
SX2	D1	3.12	3.117	0.0058	6.06	6.057	0.0058	18.8797	0.0393
	D2	3.11			6.05				
	D3	3.12			6.06				
SX3	D1	3.12	3.123	0.0058	6.08	6.073	0.0058	18.9660	0.0394
	D2	3.12			6.07				
	D3	3.13			6.07				
SX4	D1	3.12	3.123	0.0058	6.08	6.080	0	18.9878	0.3510
	D2	3.12			6.08				
	D3	3.13			6.08				
SX5	D1	3.05	3.057	0.0058	6.04	6.043	0.0058	18.4735	0.0391
	D2	3.06			6.05				
	D3	3.06			6.04				

**Table 3 materials-18-03808-t003:** Results of cross-sectional area measurements of the tested part of the SY samples.

Test Sample	Measuring Point	Thickness [mm]	Average [mm]	Standard Deviation [mm]	Width [mm]	Average [mm]	Standard Deviation [mm]	Cross-Sectional Area [mm^2^]	Standard Deviation [mm^2^]
SY1	D1	3.02	3.013	0.0058	6.04	6.033	0.0058	18.1774	0.0389
	D2	3.01			6.03				
	D3	3.01			6.03				
SY2	D1	3.07	3.067	0.0058	6.04	6.040	0	18.5247	0.0349
	D2	3.06			6.04				
	D3	3.07			6.04				
SY3	D1	3.08	3.077	0.0058	6.09	6.087	0.0058	18.7297	0.0394
	D2	3.08			6.08				
	D3	3.07			6.09				
SY4	D1	3.10	3.090	0.0100	6.06	6.063	0.0058	18.7347	0.0632
	D2	3.08			6.07				
	D3	3.09			6.06				
SY5	D1	3.02	3.023	0.0058	6.05	6.063	0.0058	18.3284	0.0391
	D2	3.03			6.05				
	D3	3.02			6.06				

**Table 4 materials-18-03808-t004:** Results of tensile tests for perpendicular samples.

Test Sample	R_eH_ [MPa]	Strain ε_ReH_ [%]	R_m_ [MPa]	Strain ε_RM_ [%]
SX1	730.25	0.41	781.48	3.49
SX2	714.98	0.39	764.44	3.70
SX3	709.26	0.38	760.42	3.54
SX4	706.42	0.43	757.75	3.57
SX5	720.89	0.38	776.10	3.68
Average	716.36	0.40	768.04	3.60
Standard deviation	9.55	0.02	10.28	0.09

**Table 5 materials-18-03808-t005:** Results of tensile tests for parallel samples.

Test Sample	R_eH_ [MPa]	Strain ε_ReH_ [%]	R_m_ [MPa]	Strain ε_Rm_ [%]
SY1	781.01	0.32	820.21	3.07
SY2	759.30	0.37	801.68	3.02
SY3	762.14	0.36	801.53	3.21
SY4	751.22	0.37	794.34	3.24
SY5	771.89	0.36	810.28	3.24
Average	765.11	0.36	805.61	3.16
Standard deviation	11.56	0.02	9.93	0.10

## Data Availability

The original contributions presented in this study are included in the article. Further inquiries can be directed to the corresponding author.

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
