# Peer review of "The Influence of Rolling Direction and Dynamic Strengthening on the Properties of Steel"

_materials, 2025, doi:10.3390/ma18163808_

Round 1
Reviewer 1 Report
Comments and Suggestions for Authors
This experimental study investigates the influence of rolling direction and strain rate on the mechanical properties of S700MC high-strength low-alloy (HSLA) steel. There are some recommendations for improvement:
- While five specimens per group are acceptable, some tables (e.g., Table 5 and 7) show slight inconsistencies in measurement precision (e.g., standard deviation of cross-sectional area jumps from 0.04 to 0.35 mm² in SX4). This may indicate minor measurement errors or surface irregularities. A brief discussion on measurement uncertainty would strengthen the analysis.
- The study infers directional strengthening from macroscopic tests but lacks microhardness mapping or EBSD (electron backscatter diffraction) data to quantify crystallographic texture or phase distribution anisotropy. Please provide some comments.
- The article briefly mentions structural uses of S700MC but primarily focuses on monotonic tensile properties. Other literature assesses weldability issues (e.g., HAZ softening, toughness drops post-welding), fatigue life after welding and in service, and the performance under cyclic or impact loading. Given the established importance of weldability, fatigue, and impact performance in structural applications of S700MC steel, situating the current findings within this broader context would enhance their practical utility for engineering design.
Author Response
Dear Reviewer,
On behalf of all authors, I would like to thank you for taking your time to read our manuscript and put your comments, which have allowed us to improve the quality of our work. Below you can find our answers related to each of your comments.
- While five specimens per group are acceptable, some tables (e.g., Table 5 and 7) show slight inconsistencies in measurement precision (e.g., standard deviation of cross-sectional area jumps from 0.04 to 0.35 mm² in SX4). This may indicate minor measurement errors or surface irregularities. A brief discussion on measurement uncertainty would strengthen the analysis.
Ad.1. We have added descriptions of measurement errors.
- The study infers directional strengthening from macroscopic tests but lacks microhardness mapping or EBSD (electron backscatter diffraction) data to quantify crystallographic texture or phase distribution anisotropy. Please provide some comments.
Ad.2. In our study, we focused on the macroscopic analysis and mechanical properties of S700MC steel with respect to rolling direction and dynamic loading. Due to the limited scope of this work, microhardness measurements and crystallographic texture analysis using the EBSD method were not included.
- The article briefly mentions structural uses of S700MC but primarily focuses on monotonic tensile properties. Other literature assesses weldability issues (e.g., HAZ softening, toughness drops post-welding), fatigue life after welding and in service, and the performance under cyclic or impact loading. Given the established importance of weldability, fatigue, and impact performance in structural applications of S700MC steel, situating the current findings within this broader context would enhance their practical utility for engineering design.
Ad.3. Thank you for your review. We agree that issues such as weldability, fatigue life, and impact resistance are crucial for the structural applications of S700MC steel, but due to equipment limitations, we were unable to expand these studies. However, we will definitely expand our future work to include these issues.
Reviewer 2 Report
Comments and Suggestions for Authors
Article title: “The Influence of Rolling Direction and Dynamic Strengthening on the Properties of Steel”
- Scientific Contribution of the Article
The main contribution of this work lies in the experimental characterization of the influence of rolling direction and dynamic strengthening on the mechanical properties of S700MC steel. Through static and dynamic tensile tests, supported by Digital Image Correlation (DIC), the authors analyze the anisotropy of the material’s mechanical behavior and its evolution under different loading conditions. - Comments on the Abstract
The abstract is clear and technically appropriate. It effectively communicates the study’s objectives, experimental approach, variables analyzed, and general conclusions. However, it adopts an overly descriptive tone and lacks a critical or quantitative focus on the results. It would be desirable to include more specific comparative data (e.g., percentage increase in strength according to specimen orientation) and a more explicit mention of the findings’ relevance for structural design applications. - Comments on the Introduction
The introduction is thorough and provides appropriate technical context regarding S700MC steel, its industrial applications, microstructure, and behavior under static and dynamic loading. Key references are cited appropriately to support the choice of material and justify the relevance of the study. However, the text tends to reiterate well-known information about HSLA steels without clearly identifying a specific knowledge gap. This weakens the scientific justification of the study, which relies more on a technical overview than on the explicit formulation of an unresolved research question. - Comments on the Applied Methodology
The authors delve into the description of the materials and experimental procedures without first presenting the general research methodology or explicitly stating the study’s objectives. Additionally, several specific concerns arise:
4.1 Lack of comparable dynamic tests: Although the influence of dynamic strengthening is mentioned, the analysis is limited to a single stress–strain curve under impact loading, with no replication or statistical treatment, preventing a robust evaluation of behavior under different strain rates.
4.2 Absence of detailed crystallographic characterization: The anisotropy due to rolling is acknowledged, but no crystallographic data (e.g., EBSD) are provided to correlate grain orientation with mechanical behavior.
4.3 Limited sample size: Only five specimens were tested for each orientation. While acceptable, this is a modest number to support generalizable claims about anisotropy and material reliability.
- Conclusions
The conclusions are well-structured and consistent with the experimental results reported. They adequately summarize the relationship between cutting orientation, strength, and ductility, and acknowledge the relevance of microstructural defects in fatigue resistance. However, the closing statements are somewhat optimistic regarding the applicability of S700MC steel, without sufficiently considering the methodological limitations previously discussed or the absence of simulations and complex loading scenarios. - Bibliographic References
The article cites a reasonable mix of recent sources (2020–2024) and applied technical studies, particularly in the context of S700MC steel. However, the references do not adequately cover constitutive models or advanced simulation works, which could have enriched the dynamic analysis and discussion. References 6 and 8 are duplicated. - Other Editorial Aspects
- In Figure 1, two projections are shown; each should be labeled (e.g., “a”, “b”) and clearly described in the caption. Units should be included.
- Table 3 contains overlapping characters and misaligned rows. It would be appropriate to adjust the table width to fit the page layout. The same applies to Table 4.
- In Figure 11, a legend should be added to explain the meaning of the curve colors.
- In Figure 12, each subfigure should be labeled (e.g., “a”, “b”, etc.). The figure caption should also describe the content of each subfigure accordingly.
Author Response
Dear Reviewer,
On behalf of all authors, I would like to thank you for taking the time to read our manuscript and for your comments, which have allowed us to improve the quality of our work. Below you can find our answers related to each of your comments.
- The main contribution of this work lies in the experimental characterization of the influence of rolling direction and dynamic strengthening on the mechanical properties of S700MC steel. Through static and dynamic tensile tests, supported by Digital Image Correlation (DIC), the authors analyze the anisotropy of the material’s mechanical behavior and its evolution under different loading conditions.
Ad.1. Thank you for your reviews. We have tried to outline the purpose of our research more precisely in the introduction and summary.
- The abstract is clear and technically appropriate. It effectively communicates the study’s objectives, experimental approach, variables analyzed, and general conclusions. However, it adopts an overly descriptive tone and lacks a critical or quantitative focus on the results. It would be desirable to include more specific comparative data (e.g., percentage increase in strength according to specimen orientation) and a more explicit mention of the findings’ relevance for structural design applications.
Ad.2. We have revised the summary to include detailed data.
- The introduction is thorough and provides appropriate technical context regarding S700MC steel, its industrial applications, microstructure, and behavior under static and dynamic loading. Key references are cited appropriately to support the choice of material and justify the relevance of the study. However, the text tends to reiterate well-known information about HSLA steels without clearly identifying a specific knowledge gap. This weakens the scientific justification of the study, which relies more on a technical overview than on the explicit formulation of an unresolved research question.
Ad.3. We have corrected the last paragraph of the introduction to more accurately emphasize the purpose of our research.
- The authors delve into the description of the materials and experimental procedures without first presenting the general research methodology or explicitly stating the study’s objectives. Additionally, several specific concerns arise:
Ad.4. We added a paragraph to section 2 describing the methodology.
- Lack of comparable dynamic tests: Although the influence of dynamic strengthening is mentioned, the analysis is limited to a single stress–strain curve under impact loading, with no replication or statistical treatment, preventing a robust evaluation of behavior under different strain rates.
Ad.5. We conducted tests on five samples, two of which we rejected. Due to the lack of relevant standards for such tests, we did this on a pilot basis.
- Absence of detailed crystallographic characterization: The anisotropy due to rolling is acknowledged, but no crystallographic data (e.g., EBSD) are provided to correlate grain orientation with mechanical behavior.
Ad.6. In our study, we focused on the macroscopic analysis and mechanical properties of S700MC steel with respect to rolling direction and dynamic loading. Due to the limited scope of this work, microhardness measurements and crystallographic texture analysis using the EBSD method were not included.
- Limited sample size: Only five specimens were tested for each orientation. While acceptable, this is a modest number to support generalizable claims about anisotropy and material reliability.
Ad.7. In accordance with ASTM E8/E8M, we used five samples for each orientation, which is considered sufficient for comparing basic mechanical properties under laboratory conditions. Nevertheless, we agree that a larger number of samples would allow for a more reliable statistical analysis and further generalizations regarding the reliability of the material. We will certainly take this into account in future studies.
- The conclusions are well-structured and consistent with the experimental results reported. They adequately summarize the relationship between cutting orientation, strength, and ductility, and acknowledge the relevance of microstructural defects in fatigue resistance. However, the closing statements are somewhat optimistic regarding the applicability of S700MC steel, without sufficiently considering the methodological limitations previously discussed or the absence of simulations and complex loading scenarios.
Ad.8. We added a paragraph to the conclusions containing these issues.
- The article cites a reasonable mix of recent sources (2020–2024) and applied technical studies, particularly in the context of S700MC steel. However, the references do not adequately cover constitutive models or advanced simulation works, which could have enriched the dynamic analysis and discussion. References 6 and 8 are duplicated.
Ad.9. We expanded this paragraph and corrected duplicate citations.
- In Figure 1, two projections are shown; each should be labeled (e.g., “a”, “b”) and clearly described in the caption. Units should be included.
Ad.10. We have changed the figure as recommended.
- Table 3 contains overlapping characters and misaligned rows. It would be appropriate to adjust the table width to fit the page layout. The same applies to Table 4.
Ad.11. We have corrected these tables.
- In Figure 11, a legend should be added to explain the meaning of the curve colors.
Ad.12. We added a legend to the chart.
- In Figure 12, each subfigure should be labeled (e.g., “a”, “b”, etc.). The figure caption should also describe the content of each subfigure accordingly.
Ad.13. We added markings and descriptions
Reviewer 3 Report
Comments and Suggestions for Authors
The article is devoted to the study of the influence of the rolling direction and dynamic hardening on the mechanical properties of high-strength low-alloy steel S700MC. The paper presents the results of experimental tests, including static and dynamic tension of samples oriented along and across the rolling direction. Particular attention is paid to the anisotropy of properties, manifested in differences in strength and ductility, as well as the behavior of the material at high strain rates. To analyze local deformations and identify zones of concentrated stress, the digital image correlation (DIC) technique is used. The fractographic analysis made it possible to establish the nature of material failure and assess the presence of defects that can initiate fatigue cracks. The work is highly relevant in the context of developing reliable structural elements for the transport, energy and construction industries. The results obtained contribute to a more accurate design of parts taking into account the rolling direction and loading conditions, which is important for increasing the reliability and service life of S700MC steel products. At the same time, the article requires a number of revisions:
1. The formulation of the research objective at the end of the introduction (lines 89–93) requires clarification. In its current form, the objective is too descriptive and does not contain a hypothesis or a forecast. It is recommended to supplement the formulation by indicating what specific changes in the mechanical properties were expected when changing the rolling direction and dynamic loading.
2. The description of the sample preparation (lines 126–157) does not indicate whether heat treatment was carried out after machining, especially after using a milling machine. Since even moderate heating can affect the structure and properties of thin sheet steel, it is advisable to describe whether measures were taken to exclude thermal influences.
3. The metallographic analysis (lines 116–125) is qualitative, but there is no quantitative assessment of the microstructure, for example, by grain size or phase composition. To confirm the homogeneity of the structure, quantitative analysis methods (for example, according to ASTM E112) should be used and histograms of the grain size distribution should be included. 4. Section 3.3 on static testing (lines 240–260) does a good job of highlighting the differences between the along- and across-mill oriented specimens. However, the discussion of the results does not take into account texture effects that could be confirmed by, for example, EBRS or diffraction. Their absence reduces the depth of the analysis.
5. Section 3.3 (lines 264–271) shows a significant increase in strength for the SY specimens, but does not discuss the reasons for this increase other than orientation. Residual texture or inclusion direction may have played a role. It is recommended that the discussion be supplemented with possible microstructural mechanisms.
6. On lines 291–299, the authors demonstrate a discrepancy between the traditional stress-strain curve and the DIC data. However, this discrepancy is discussed only in terms of necking, without a numerical comparison of the real and engineering stress. The reviewer recommends including a comparison of the true stress-strain curves. 7. The description of the dynamic test results (lines 331–344) lacks a comparison between the static and dynamic behavior of the material. It is recommended to provide curves or tables with comparable strength characteristics to show the effect of dynamic hardening.
8. The fractographic analysis (lines 347–384) is presented in detail, but requires comparison with literature data. It would be useful to include a comparison with the microstructures of other HSLA steels and to discuss how typical the observed dimples and internal pores are.
9. The conclusions (lines 389–421) provide reasonable conclusions, but some of them duplicate the description of the results instead of generalizing. It is recommended to restate the conclusions in terms of engineering recommendations or applied implications for designers.
10. The point about the significance of microdefects in the conclusion (lines 410–412) deserves separate discussion. It is necessary to clarify how it is proposed to minimize such defects - by improving rolling conditions, heat treatment or other methods?
Author Response
Dear Reviewer,
On behalf of all authors, I would like to thank you for taking the time to read our manuscript and for your comments, which allowed us to improve the quality of our work. Below you can find our answers related to each of your comments.
- The formulation of the research objective at the end of the introduction (lines 89–93) requires clarification. In its current form, the objective is too descriptive and does not contain a hypothesis or a forecast. It is recommended to supplement the formulation by indicating what specific changes in the mechanical properties were expected when changing the rolling direction and dynamic loading.
Ad.1. The research objective and the research hypothesis are presented in lines 100–112.
- The description of the sample preparation (lines 126–157) does not indicate whether heat treatment was carried out after machining, especially after using a milling machine. Since even moderate heating can affect the structure and properties of thin sheet steel, it is advisable to describe whether measures were taken to exclude thermal influences.
Ad.2. The sample preparation process was carried out using a CNC machine tool equipped with a coolant supply system. Moreover, after the machining process, the samples were manually removed, during which the material temperature allowed them to be safely handled. This fact eliminates the possibility of high material temperature. Nevertheless, no additional heat treatment was performed.
- The metallographic analysis (lines 116–125) is qualitative, but there is no quantitative assessment of the microstructure, for example, by grain size or phase composition. To confirm the homogeneity of the structure, quantitative analysis methods (for example, according to ASTM E112) should be used and histograms of the grain size distribution should be included.
Ad.3. In the present study, measurements of grain size or phase composition were not performed. Initially, the analysis of the material’s structure was based on the technical documentation and specifications provided by the manufacturer. Only after this preliminary verification was a microscopic examination conducted in order to observe the general features of the microstructure. However, this analysis was limited to a qualitative evaluation, without quantifying grain size, grain size distribution, or phase fractions. We agree that incorporating quantitative metallographic data—such as grain size measurements according to ASTM E112 and corresponding histograms—would strengthen the assessment of structural homogeneity. This is a valuable suggestion, which we will take into consideration in future work.
- Section 3.3 on static testing (lines 240–260) does a good job of highlighting the differences between the along- and across-mill oriented specimens. However, the discussion of the results does not take into account texture effects that could be confirmed by, for example, EBRS or diffraction. Their absence reduces the depth of the analysis.
Ad. 4. In section 3.3 we focused primarily on the mechanical differences between specimens oriented along and across the rolling direction. However, we did not perform texture analysis using methods such as EBSD or X-ray diffraction, and therefore the influence of crystallographic texture on the observed results could not be directly assessed. The primary aim of this section was to compare the mechanical behavior in different orientations based on macroscopic testing. While we agree that texture could significantly influence the mechanical anisotropy, such analyses were beyond the scope of the current study. Nevertheless, we fully acknowledge the value of such an approach, and we intend to include texture characterization in future investigations to provide a more comprehensive interpretation of the material behavior. - Section 3.3 (lines 264–271) shows a significant increase in strength for the SY specimens, but does not discuss the reasons for this increase other than orientation. Residual texture or inclusion direction may have played a role. It is recommended that the discussion be supplemented with possible microstructural mechanisms.
Ad.5. Due to the limitations of our study, we did not conduct a detailed analysis of residual texture or inclusion alignment. Nonetheless, we recognize that such microstructural features could significantly influence the mechanical response. Residual crystallographic texture, inherited from the rolling process, the alignment of non-metallic inclusions, and elongated grains parallel to the rolling direction may all contribute to the improved strength observed in the SY specimens. In response to the reviewer’s suggestion, we have supplemented the discussion with a paragraph explaining these possible microstructural mechanisms. This addition can be found in lines 336–344 of the revised manuscript.
- On lines 291–299, the authors demonstrate a discrepancy between the traditional stress-strain curve and the DIC data. However, this discrepancy is discussed only in terms of necking, without a numerical comparison of the real and engineering stress. The reviewer recommends including a comparison of the true stress-strain curves.
Ad. 6. The discrepancy between the traditional engineering stress–strain curve and the DIC-based observation was originally discussed only in the context of necking. We agree that a numerical comparison with true stress–strain curves would provide additional depth to the analysis. However, due to the limitations of our experimental setup, only qualitative data from DIC in the form of deformation images were available. The system was not configured to extract full-field numerical strain data that would allow for the calculation of true stress–strain curves. Therefore, a direct numerical comparison between engineering and true stress could not be performed. In response to the reviewer’s suggestion, we have clarified this limitation in the revised manuscript and added a qualitative explanation of how the DIC data support the expected behavior of true stress beyond the onset of necking. This information has been incorporated in lines 400–407.
- The description of the dynamic test results (lines 331–344) lacks a comparison between the static and dynamic behavior of the material. It is recommended to provide curves or tables with comparable strength characteristics to show the effect of dynamic hardening.
Ad.7. In response, we have revised the manuscript to include a direct comparison between the dynamic and static tensile behavior of the tested steel. Specifically, we have added a new figure (Fig. 11) that presents the engineering stress–strain curves for all three dynamic test replicates (D1, D2, D3) alongside the quasi-static reference sample (SY1). - The fractographic analysis (lines 347–384) is presented in detail, but requires comparison with literature data. It would be useful to include a comparison with the microstructures of other HSLA steels and to discuss how typical the observed dimples and internal pores are.
Ad.8. In the revised manuscript, we have added a comparative discussion referencing the typical fracture surface morphology observed in other HSLA steels, including S500MC and S960QC. The presence of microvoids and dimples, as observed in the tested S700MC steel, is a well-documented feature of ductile fracture in fine-grained HSLA steels, particularly when failure occurs in the necking region under triaxial stress states.
- The conclusions (lines 389–421) provide reasonable conclusions, but some of them duplicate the description of the results instead of generalizing. It is recommended to restate the conclusions in terms of engineering recommendations or applied implications for designers.
Ad.9. Thank you for this suggestion. We have revised the Conclusions section to shift the emphasis from a descriptive summary of the test results to a more application-oriented format. Specifically, we introduced recommendations for designers and engineers regarding the selection of rolling direction and consideration of strain rate effects in structural applications. The updated conclusions highlight how S700MC’s directional mechanical behavior and strain-rate sensitivity should be factored into material selection and orientation in components subjected to dynamic or crash-relevant loading. We have also commented on the importance of defect control in improving fatigue life and structural integrity.
- The point about the significance of microdefects in the conclusion (lines 410–412) deserves separate discussion. It is necessary to clarify how it is proposed to minimize such defects - by improving rolling conditions, heat treatment or other methods?
Ad. 10. We appreciate this observation. In response, we have expanded the discussion to clarify the potential origins of internal defects and provide practical approaches to minimize their presence in industrial production. The revised text emphasizes that such microdefects likely originate from rolling-induced segregation, non-metallic inclusions, or improper coiling temperatures.
Round 2
Reviewer 2 Report
Comments and Suggestions for Authors
I have no further comments.
Author Response
Thank you very much
Reviewer 3 Report
Comments and Suggestions for Authors
The authors have substantially revised the article — the structure has been improved, additional explanations and comparative materials have been introduced, and the conclusions have been revised in an applied vein. However, a number of comments related to in-depth quantitative analysis of the microstructure and texture, numerical comparison of true stress curves, and more detailed comparison with literature data have remained unimplemented.
What has not been fully implemented:
Quantitative metallographic analysis (grain size, distribution, phase composition) according to ASTM E112 has not been performed; only a qualitative assessment remains (lines 243–255).
Comparison of the results with literature data on dynamic hardening has been limited — mainly at the level of fractography, but without statistical or parametric linking.
Author Response
Dear Reviewer,
On behalf of all authors, I would like to thank you for your additional comments. Below you can find our answers.
The authors have substantially revised the article — the structure has been improved, additional explanations and comparative materials have been introduced, and the conclusions have been revised in an applied vein. However, a number of comments related to in-depth quantitative analysis of the microstructure and texture, numerical comparison of true stress curves, and more detailed comparison with literature data have remained unimplemented. What has not been fully implemented:
- Quantitative metallographic analysis (grain size, distribution, phase composition) according to ASTM E112 has not been performed; only a qualitative assessment remains (lines 243–255).
Ad.1. We acknowledge the reviewer’s observation. As stated in the revised manuscript, the present study relied on qualitative metallographic assessment supported by manufacturer documentation. Due to the scope of the current work and limitations in available experimental resources, quantitative metallographic analysis according to ASTM E112 (including grain size measurements, distribution histograms, and phase composition) was not conducted. We agree that such analysis would significantly strengthen the interpretation of microstructural homogeneity and anisotropy. This limitation has now been explicitly stated in lines 249–254 of the revised manuscript. Such analysis is planned in a follow-up study where these parameters will be correlated with mechanical properties under both static and dynamic loading conditions.
- Comparison of the results with literature data on dynamic hardening has been limited — mainly at the level of fractography, but without statistical or parametric linking.
Ad.2. We thank the reviewer for this important comment. In the revised manuscript, we have expanded the discussion of dynamic hardening to include a direct numerical comparison between static and dynamic tensile test results, as well as a broader literature context. For the tested S700MC steel, dynamic loading increased the ultimate tensile strength (Rm) by approximately 19% compared to static conditions. This result is consistent with strain-rate sensitivity values reported for other HSLA steels, where increases of 6–20% in Rm have been observed at comparable strain rates. We have also described the underlying strengthening mechanisms, including reduced dynamic recovery time for dislocations and the influence of microstructural features such as grain refinement and precipitate distribution. This expanded comparison and discussion provide both numerical and mechanistic linkage between our findings and published data, addressing the reviewer’s request for a more in-depth parametric connection. The new content has been added in lines 431–446 of the revised manuscript.